# High-Intensity Interval Training Combined with Different Types of Exercises on Cardiac Autonomic Function—An Analytical Cross-Sectional Study in CrossFit^®^ Athletes

**DOI:** 10.3390/ijerph20010634

**Published:** 2022-12-30

**Authors:** Michelle Teles Morlin, Carlos Janssen Gomes da Cruz, Freddy Enrique Ramos Guimarães, Renato André Sousa da Silva, Luiz Guilherme Grossi Porto, Guilherme Eckhardt Molina

**Affiliations:** 1Exercise Physiology Laboratory, Faculty of Physical Education, University of Brasilia, Brasilia 70910-900, DF, Brazil; 2SARAH Network of Rehabilitation Hospitals, Brasilia 70910-900, DF, Brazil; 3GEAFS—Research Group on Physiology and Epidemiology of Exercise and Physical Activity, Brasilia 70910-900, DF, Brazil; 4Laboratory of Physical Performance and Healthy, Faculty of Physical Education, Euro American University Center, Brasilia 70200-001, DF, Brazil; 5Goiano Federal Institute, Campus Morrinhos, Morrinhos 75650-000, GO, Brazil

**Keywords:** autonomic nervous system, parasympathetic, athletes, high-intensity interval training, exercise

## Abstract

It is well established that endurance exercise has positive effects on cardiac autonomic function (CAF). However, there is still a dearth of information about the effects of regular high-intensity interval training combined with different types of exercises (HIITCE) on CAF. Objective: The aim of this study is to compare CAF at rest, its reactivity, and reactivation following maximal exercise testing in HIITCE and endurance athletes. Methods: An observational study was conducted with 34 male athletes of HIITCE (i.e., CrossFit^®^) [HG: n = 18; 30.6 ± 4.8 years] and endurance athletes (i.e., triathlon) [TG.: n = 16; 32.8 ± 3.6 years]. We analyzed 5 min of frequency-domain indices (TP, LF, HF, LFn, HFn, and LF/HF ratio) of heart rate variability (HRV) in both supine and orthostatic positions and its reactivity after the active orthostatic test. Post-exercise heart rate recovery (HRR) was assessed at 60, 180, and 300 s. Statistical analysis employed a non-parametric test with a *p*-value set at 5%. Results: The HG showed reduced HFn and increased LFn modulations at rest (supine). Overall cardiac autonomic modulation (TP) at supine and all indices of HRV at the orthostatic position were similar between groups. Following the orthostatic test, the HG showed low reactivity for all HRV indices compared to TG. After the exercise, HRR does not show a difference between groups at 60 s. However, at 180 and 300 s, an impairment of HRR was observed in HG than in TG. Conclusion: At rest (supine), the HG showed reduced parasympathetic and increased sympathetic modulation, low reactivity after postural change, and impaired HRR compared to TG.

## 1. Introduction

Physical inactivity is one of the major risk factors for developing cardiovascular disease and is strongly associated with an increased risk of cardiovascular mortality [1]. The cardiac autonomic function (CAF) is associated with physical inactivity, and its impairment is a predictor of all-cause and cardiovascular disease mortality [2].

Frequency-domain analysis of spontaneous heart rate variability (HRV) based on the oscillation in the intervals between consecutive heartbeats (R-R intervals) is a suitable xmethod for assessing the relative modulation of parasympathetic and sympathovagal balance on the sinus node [3].

In the clinical setting, HRV response to gravitational stress imposed by active postural change (orthostatic test) is used for assessing the cardiac autonomic reactivity, which shows the shift in the autonomic balance by reducing parasympathetic activity and enhancing sympathetic activity in different degrees according to the resting activity status [4,5].

In the condition of post-exercise, heart rate recovery (HRR) is another feasible tool for evaluating CAF [6]. HRR is defined as the rate at which heart rate (HR) decreases within the following minutes after exercise and displays the dynamic balance and coordinated interplay between parasympathetic reactivation and sympathetic deactivation [6,7]. Hence, attenuated HRR is associated with an increased risk of cardiovascular events and all-cause mortality [7].

Over the years, studies have shown that aerobic training has positive clinical effects, such as the prevention of or decrease in the incidence of cardiovascular problems and positive changes in CAF [8,9,10,11,12], improving HRV (modulation and reactivity) and HRR. Thus, regular aerobic exercise training is widely recommended to reduce cardiovascular morbidity and all-cause mortality [10,11,12,13,14].

A recent worldwide survey showed that high-intensity exercise methods have been considered the leading trend in the fitness business world [15]. In this context, and based on a recent discussion about exercise concepts, CrossFit^®^ is recognized as high-intensity interval training combined with different types of exercises (HIITCE) [16]. CrossFit^®^ programs are developed to address multiple fitness domains (combined aerobic and resistance exercises), potentially improving physical health [17], and are considered an exercise option to promote health [18,19].

Some studies have shown the potential high injury risk of CrossFit^®^ [20,21,22]. Recently, a systematic review has shown a positive influence on body composition and physical fitness [17] in CrossFit^®^ practitioners. However, there is still a dearth of information about the impact of HIITCE (i.e., CrossFit^®^) on CAF in individuals who practice this modality exclusively. In addition, to our knowledge, only two articles described the CAF in CrossFit^®^ practitioners, but none have compared the HIITCE with another sport modality [23,24]. Hence, considering the positive effect of regular, continuous aerobic training (i.e., triathlon) on CAF, as described above, and the scarcity of studies showing the impact of CrossFit^®^ on CAF and the worldwide interest in this modality, it becomes essential to describe and analyze the CAF (modulation, reactivity, and reactivation) in CrossFit^®^ athletes and compare these two distinct modalities. One is characterized by the high volume of continuous training demand, and the other by high-intensity interval training combined with different types of exercise demand on CAF.

Therefore, given the scarcity of studies showing the effects of CrossFit^®^ on CAF associated with growing worldwide interest in this modality and its potential health benefits, this study aimed to compare the CAF at rest, its reactivity, and reactivation following the maximal treadmill exercise testing in healthy male CrossFit^®^ athletes comparatively to healthy male triathlon athletes.

## 2. Materials and Methods

### 2.1. Subjects

This observational study was composed of convenience sampling with thirty-four men (n = 34). Participants were eligible for inclusion if they were men, healthy, aged between 20 and 40 years old, with at least one year of regular exercise routine. They were previously instructed to refrain from stimulants and alcoholic beverages, medicines, and physical activity for 48 h prior to the tests. The local ethics committee approved the experimental protocol (n° 1.151.967), and all participants signed informed written consent to participate in this study. The participants were split into two groups: the triathlon group (TG, n = 16), composed of recreational triathlon athletes, and the HIITCE group (HG, n = 18), composed of recreational CrossFit^®^ practitioners, e.g., open competitors. All of the HIITCE group was derived from the true nature of the CrossFit^®^ brand, including the gym (affiliate map on CrossFit’s official homepage) with official CrossFit classes and certified CrossFit coaches (at least level 1 certificate course and physical education graduation).

The data were collected in December and January 2018/2019, corresponding to athletes’ training maintenance phase and vacation. None of the athletes were in a specific training or pre-competition phase. Before the tests, the athletes were instructed to rest and stop physical training for 48 h. It is important to emphasize that the maintenance period was crucial for the athletes to perform 48 h of rest, or adherence would be low if the data collection were carried out during the specific season or pre-competition phase. Regarding training, the triathlon athletes performed endurance activities in an average time of 4 years daily. During the data collection, the routine consisted of an average of 2 daily sessions of aerobic activity, 7/6 days a week, totaling an average of 17 h per week with moderate intensity; CrossFit^®^ athletes (RX classification) performed activities in an average time of 3.3 years. During the data collection, the routine consisted of 1 daily session of activities ranging from endurance, strength training, and calisthenics in an interval and high-intensity exercise, 6 days a week, totaling an average of 7 h per week. No athlete included in the study reported feelings of fatigue, sleep disturbances, apathy, or restlessness during the days before data collection. In addition, a cardiologist evaluated all athletes to verify the same clinical health condition and no signs of overtraining or overreaching.

Therefore, the clinical examination was conducted by a cardiologist with a 12-lead electrocardiograph in the supine position. Then, anthropometrical and basic physiological data (heart rate (HR), blood pressure, and respiratory rate) were collected by the same principal researcher, besides information on lifestyle habits. The anthropometrical and basic physiological characteristics are presented in Table 1. The HRV examination was performed in the supine position in a quiet exercise physiology laboratory room between 8:00 and 10:00 a.m., at ambient temperature (21–24 °C) and relative humidity of 50 to 60%. After 15 min of rest in the supine position, a valid 5 min *R-R* interval series was recorded. Afterward, the subjects were asked to actively adopt the orthostatic posture (change from supine to orthostatic position) at the bedside. After 2 min in this position, an additional 5 min *R-R* interval series was recorded [25]. Before the second *R-R* intervals recording, blood pressure was measured to verify the absence of postural hypotension.

The orthostatic stress test was made actively and without any support to the subjects with the per-protocol instruction to achieve the new body position in no more than 10 s [5,26].

Throughout the *R-R* intervals series recording, the subjects had their respiratory rate (RR) observed to ensure they were above nine breaths per minute. Such a procedure is necessary for the frequency-domain HRV analysis as it avoids overlapping the low- and high-frequency spectral band areas [27]. The subjects breathed spontaneously, had their respiratory rate visually monitored, and counted the number of chest expansions and retractions for 1 min during the R-R intervals recording [28]. Participants with respiratory rates below nine breaths per minute and who presented altered emotional state (self-declared) were excluded from the study.

Soon after the orthostatic stress test, a maximum treadmill incremental exercise stress test was applied, and the participants proceeded with the post-exercise active cool-down period [7].

### 2.2. Heart Rate Variability

The *R-R* intervals series were recorded using a valid and reliable heart rate monitor Polar^®^ (model RS800CX, Kempele, Finland, [29]. Afterward, each series was transferred to a microcomputer for offline data processing and analysis of *R-R* intervals variability, employing MATLAB-ECGLAB software(Cardiovascular Laboratory/FM-UNB, Brasília, Brazil) [29,30].

Before the HRV analysis, each *R-R* intervals series was initially checked beat-to-beat for confirmation of sinus rhythm, identification of non-sinus and ectopic beats, artifacts, and signal stability. The series from which ectopic or other non-sinusal beats and outliers were removed was submitted to interpolation by cubic splines method, and then processed and analyzed [30].

As the *R-R* interval series were recorded in strictly controlled and stable experimental conditions, those qualified for analysis were highly stationary, as estimated by the percent differences of the mean and the standard deviation between each pair of 3 segments of the *R-R* intervals series. The series was processed and analyzed for variability in the frequency domain employing different conventional indices.

For the spectral analysis, the series were normalized and resampled at 4 Hz using the cubic polynomial interpolation method (cubic spline) to make the intervals equidistant and continuous. Then, they were filtered by Hanning windowing to attenuate the discontinuity effects, and then they were processed by the 16-order autoregressive modeling to convert the wave components into a frequency spectrum, which comprises very-low-frequency (VLF; 0–0.04 Hz), low-frequency (LF; 0.04–0.15 Hz), and high-frequency (HF; 0.15–0.50 Hz) spectral bands [30].

Frequency-domain indices calculated included: (a) total power spectral area (00.50 Hz) (TP), which indicates the overall autonomic activity; (b) absolute power areas of the low- (LF) and high-frequency (HF) bands; and (c) normalized power area of low-frequency (LFn) and high-frequency (HFn) bands, which were the percentage of absolute power area of each band with the sum of both absolute areas. Low- and high-frequency bands are, respectively, surrogates of combined sympathetic plus parasympathetic and exclusive parasympathetic activities, i.e., (d) ratio of low–high frequency absolute areas (LF/HF) that estimates the sympathovagal balance [26,31].

### 2.3. Cardiopulmonary Exercise Test and Heart Rate Recovery

According to the individualized ramp protocol, the maximal exercise stress test consisted of an incremental treadmill exercise test [32]. All tests were performed between 8 and 12 min on a conventional treadmill (ATL, Imbrasport).

The protocol started from a speed of 4km/h and at a 2.5% grade slope; this grade remained constant over the test, and the treadmill speed was increased gradually by 1.2 km/h minutes before volitional fatigue set in. Peak oxygen uptake (VO_2_peak) was determined as previously described [33] and measured by pulmonary gas exchange using the Ergoespirometer—Cortex Metalyzer 3B (Biophysik, Leipzig, Germany). The following criteria confirmed the VO_2_peak: (a) VO_2_ leveling off; (b) RERmax ≥ 1.0, Borg Scale 6–20 rating; and (c) HRmax ≥ 95% of the age-predicted HRmax (220– age) compared with symptom-limiting termination of the test.

Immediately after the maximal exercise test, the participants were engaged in post-exercise active recovery with speed reduced to 2.4 km/h, a 2.5% grade slope, as Cole et al. [7] described. This recovery protocol was adopted considering its technical feasibility, reproducibility, and frequent application in clinical and functional settings.

During the active recovery, the absolute (Δ) and relative (%Δ) values of HRR were obtained by reducing from the maximal HR (HRmax) reached during the maximal treadmill exercise test to HR at 60, 180, and 300 s throughout the active recovery phase [34,35].

### 2.4. Statistical Analysis

The variables presented a non-normal distribution (Shapiro–Wilk tests). Thus, statistical analysis uniformly employed non-parametric tests, and the variables were presented as median and quartile values.

The Mann–Whitney U-Test was used for comparison between groups: (a) the HRV indices in supine and orthostatic position; (b) absolute (Δ) and relative (%Δ) changes in HRV indices following the orthostatic stress test; and (c) the absolute and relative values of HRR throughout the post-exercise active recovery phase.

The difference level was set as a two-tailed *p*-value ≤ 0.05. The effect size (*ES*) of comparative analysis was calculated using the formula: $ES=\frac{Z}{\surd n}$ where “Z” is the z-score converted from the probability value of a test statistic and “√n” is the square root of the total sample on which “Z” is based [36,37]. According to Cohen [38], effect sizes with 0.2–0.3 were classified as small, 0.5–0.8 as medium, and ≥0.8 as large. The observed power was calculated by using post-hoc power analysis for each hypothesis via G*power 3.1.9.7 for Windows software [39].

Statistical analysis employed the Statistical Package Social Sciences (SPSS 22) and the Prism 4 for Windows (GraphPad Software, Inc., San Diego, CA, USA, 2005) software packages.

## 3. Results

The median BMI (*p* = 0.01) was significantly higher in the HG, and the training volume was significantly lower in the same group (*p* < 0.01). The HR in the supine position was statistically lower in the TG (*p* = 0.01), and the VO_2_ peak was significantly higher in the same group (*p* < 0.01). On the other hand, there was no statistical difference for all other variables analyzed between groups before and during the exercise (*p* = 0.11–0.74) (Table 1).

Although bradycardia (*R-R* intervals) was observed in both groups, the increased median *R-R* intervals in the TG were significantly higher at resting supine (*p* < 0.01; *ES*: 0.46) (Table 2). 

Regarding the HRV, in the same postural position, the HG showed significantly higher activity for the LF (*p* = 0.01; *ES*: 0.42), LFn (*p*< 0.01; *ES*: 0.41), and LF/HF ratio (*p* < 0.01; *ES*: 0.42). Regarding parasympathetic activity (HF normalized area), the median of HG showed significantly lower values (*p* < 0.01; *ES*:0.12) in the supine position (Table 2).

In the orthostatic position, all indices of HRV do not show a significant difference between groups (*p* = 0.09–0.89) (Table 2).

Concerning the absolute and relative variation from the supine to the orthostatic position (autonomic reactivity), we observed significantly reduced absolute (*p* = 0.02; *ES*: 0.38) and relative (*p* = 0.01; *ES*: 0.46) response of LF in the HG following the orthostatic stress test. The LFn showed a significantly low absolute (*p* = 0.01; *ES*: 0.31) and relative (*p* = 0.01; *ES*: 0.40) variation (increase) in the HG after postural change. Results showed the same for LF/HF ratio, in which a significantly reduced absolute and relative variation was observed in the HG (*p* = 0.01; *ES*: 0.46) and (*p* = 0.01; *ES*: 0.50), respectively. Referring to the HFn, the results showed significantly high absolute and relative variation (reduction) in TG (Table 3). The observed power of the variables related to sympathetic and parasympathetic function was observed between 0.86 and 1.

Concerning the HRR analysis following the exercise between groups, the results showed no differences between groups at 60 s of post-exercise recovery. However, at 180 and 300 s, faster HHR and ∆% HRR was observed in the TG (*p* < 0.01; ES: 0.58) and (*p* < 0.01; ES: 0.44), respectively, than in HG (Figure 1).

## 4. Discussion

We observed new and relevant findings in the present study regarding the analysis of spontaneous HRV in the frequency domain at rest (supine and orthostatic positions), its reactivity after the active postural change, and the cardiac autonomic reactivation evaluated by HRR after maximal exercise testing in HG compared to the TG. We observed that HG at rest, in the supine position, suggested a consistently sympathetic increase (enhanced absolute and normalized low-frequency power) and reduced parasympathetic modulation (reduced normalized high-frequency power). The overall autonomic activity was similar within groups, but the sympathovagal balance showed sympathetic predominance or a reduced parasympathetic modulation compared to TG in the supine position.

In the orthostatic position, we did not observe any significant difference in the cardiac autonomic modulation between groups. However, after the orthostatic stress test, the HG demonstrated low reactivity of neural modulation for both combined sympathetic plus parasympathetic and exclusive parasympathetic activities (enhanced low-frequency normalized area and sympathovagal balance toward sympathetic dominance) and reduced high-frequency normalized area after the postural change compared to TG.

Following the maximal exercise testing, the HRR at 60 s was similar between groups; however, the HG demonstrated impairment of HHR at 180 and 300 s throughout the recovery phase compared to TG.

In addition, our results showed sinus bradycardia and high levels of maximal oxygen uptake independently of the mode of exercise training (HG vs. TG). However, the highest values of sinus bradycardia and maximal oxygen uptake were observed in TG.

Therefore, as to the nature of cardiac autonomic function observed between groups, the main finding suggests that, in this study, the individuals who practice HIITCE (i.e., CrossFit^®^) as a main mode of exercise exhibited a reduction of HRV activity at rest (supine), low HRV reactivity after the postural change, and reduced reactivation through HRR during the recovery phase comparatively to TG.

Hence, the functional basis of differences in cardiac autonomic responses between groups is not a simple phenomenon to be fully understood, and its entire picture can be only conjectured, considering the complexity of all mechanisms involved in heart-rate dynamics.

One possibility is that HIITCE, as the main mode of exercise training, can induce a consistent sympathetic increase state in their practitioners, as observed in the present study. This observation can be supported by (a) high values of LF activity and reduced HF activity at the supine position; (b) reduction of absolute LF activity after the postural change in HG where the TG increases the LF power after the postural change (physiological response); and (c) the cardiac autonomic balance (LF/HF) shows lower variation in HG than TG which suggests lower reactivity of the cardiovascular system to postural changes.

Our results align with other studies that reported a shift in HRV analysis at rest towards a sympathetic predominance in athletes with overtraining syndrome (at least in the initial stage) and after acute heavy training [40,41]. In addition, athletes who are submitted to high-intensity endurance exercise showed a positive association between different fatigue states with increased LF power at rest supine and reduced LF and HF powers after the postural change [42], as observed in the present study.

The authors highlighted that the fatigue states induced by the exercise might represent a hypertonicity in sympathetic activity at rest and a hypotonicity of sympathetic and parasympathetic branches after the postural change [42].

Yet, considering the fatigue state, another hypothesis for our results is that 48 h of rest could be insufficient to settle the cardiac autonomic status in HIITCE athletes. This hypothesis is supported by studies that demonstrated that high-intensity intermittent exercise increases sympathetic predominance and affects cardiac autonomic modulation, reducing the recovery/reactivity as a function of time [9,43].

Thus, it is important to consider that autonomic activity may differ between the sports modalities in sympathetic predominance [44]. This autonomic condition could be associated with the type of sport, intensity distribution (i.e., a characteristic feature of CrossFit^®^ programs), phase of training, and also with athletes’ health status and psychological load [45,46], as often observed in endurance athletes who developed the overreaching and overtraining syndrome after submitting to high volume training and after high-intensity sections [47,48,49].

Some studies have raised concerns about a tendency to develop symptoms of overtraining in HIITCE athletes [50]. Timón et al. [51] analyzed biochemical parameters and recovery after two intensity protocols of HIITCE. In both, significant increases in hepatic transaminases, creatine phosphokinase, and subsequent decreases in physical performance were evaluated by the plank test. In a similar direction, Tibana et al. [52] demonstrated that a session of HIITCE demands a longer metabolic and hormonal recovery period than the submaximal protocol, including an increase in the concentrations of serum brain neurotrophic factor. Jacob et al. [53] emphasized that these systemic responses are related to metabolic stress caused by the high intensity combined with the variability of stimuli in this type of physical exercise.

Another possible explanation for the autonomic difference between groups would be arterial stiffness and autonomic cardiac function mediated by baroreflex [54]. Paula et al. [55] reported the delay in baroreflex response across different modalities, and arterial stiffness appears to be associated [56]. A recent study has shown that endurance training positively affects arterial stiffness [57]. On the other hand, studies have shown no changes in arterial stiffness after resistance exercise [58,59,60]. HIIT studies may positively affect arterial stiffness [61,62]; however, this is controversial [63]. In 2022, a review recently highlighted that the difference in exercise benefits might depend on the participants’ training mode, intensity, duration, and age [64]. Therefore, the benefits of interval exercise in improving autonomic and arterial stiffness still deserve investigation. Thus, as CrossFit uses high-intensity endurance, resistance, and calisthenics exercises, studies that evaluated the effect of this type of exercise on arterial stiffness in healthy adults were not found. Thus, it is still an open, incompletely explored issue.

To the best of our knowledge, our study is the first to show that those who practice HIITCE (i.e., CrossFit^®^ athletes) as the only exercise practice exhibited reduced HRV activity, reactivity, and autonomic reactivation even with a low volume of training/week compared to TG. Therefore, these results suggest exclusive HIITCE training (i.e., CrossFit^®^ athletes) may reduce cardiac autonomic function even with weekly low-volume training compared to TG.

Regarding HRR, it is well established that during incremental exercise testing, HR raises in response to parasympathetic deactivation and sympathetic activation [6,7,65]. In contrast, the short-term HR adaptation during recovery responds to simultaneous rapid parasympathetic reactivation and progressive sympathetic deactivation [6]. Therefore, our results showed that, independently of the training mode, both groups showed similar HHR at 60 s. This result has an important prognostic value from a clinical perspective because the HRR at the 1st minute of recovery is established in the literature as the standardized time for evaluating cardiovascular diseases and parasympathetic reactivity [66].

On the other hand, we observed an impaired HHR at 180 and 300 s after the exercise in the HG. Thus, the differences between HG and TG in a longer recovery period might be an important functional characteristic of both exercise modalities. The physiological mechanisms that might explain this result are beyond the purpose of the present investigation, but some potential mechanisms could be considered.

Therefore, the impairment of HRR (180 and 300 s) observed in the HG may be explained by temporary reduced parasympathetic and lower sympathetic deactivation during the recovery phase due to thermoregulation and the gradual clearance of metabolites and catecholamines released during exercise [67]. So, the impairment of HHR observed during the slow recovery phase (from 180 to 300 s) suggests that the HG displays a continuous sympathetic predominance status throughout the recovery. So, our findings indicate that future studies are needed to confirm this hypothesis.

Therefore, we believe that our results add important information regarding the worldwide interest in high-intensity interval training. Additionally, our findings must be highlighted in a scenario of the simultaneous growing popularity of high-intensity physical activities, such as CrossFit^®^, and the scarcity of studies that analyze HIITCE as the main mode of practice in overall cardiovascular health and the cardiac autonomic function during distinct functional conditions. From the point of view of cardiac autonomic function, the results showed a possible continuous sympathetic increase and reduction of parasympathetic modulation in HG compared to TG in all measurements except at the orthostatic position.

Hence, from a practical perspective, even though the HIITCE athletes displayed bradycardia, high oxygen uptake peak, and good parasympathetic reactivation during the 1st minute of post-exercise recovery, the autonomic heart control facing stimulus (acute response) showed impairment in cardiac autonomic in all analyses (rest, reactivity, and reactivation) when compared to TG. So, we believe that our data support the recommendation that the study of cardiac autonomic function by HRV should go beyond classical resting HRV indices, which includes evaluation in both the resting supine and orthostatic positions (cardiac autonomic activity), and should include the cardiac autonomic reactivity and reactivity (i.e., the active orthostatic test or post-exercise recovery). When applied, it may bring essential and complementary information related to the athlete’s cardiac autonomic status which can give a good indication of individual self-regulation. This reinforces the importance of careful assessment of the athlete’s cardiac autonomic status since reduced parasympathetic activity and increased sympathetic activity alters cardiac electrophysiological properties and seems to be related to the risk of both atrial and ventricular arrhythmias, which may pose a risk to the practice of physical activity analyzed in the present study [65,68]. In addition, the assessment of comprehensive cardiac autonomic analysis (rest, reactivity, and reactivation) can provide coaches with greater knowledge in exercise prescription as a preliminary tool for decision making (i.e., training control) and also can provide advances in sports medical prescription during rehabilitation [69] with greater knowledge related to the recovery of the organism and overreaching/overtraining status [47,48,49,70].

However, we highlight some limitations of the current study, such as the cross-sectional observational design, the absence of a sedentary control group, a non-random sample, and the lack of individual control of training sessions associated with checking signs of overtraining or overreaching. Hence, the present study does not allow a claim that the cardiac autonomic response observed in the HG during the experiment (activity, reactivity, and reactivation) was a response that starts from a chronic autonomic adaptation state of HIITCE (i.e., characteristic of this modality). On the other hand, from the physiological perspective, our results should be carefully observed. Indeed, independent of the cardiac autonomic adaptation state throughout the experiment, even discrete impairment in cardiac autonomic adjustments to distinct functional stimuli (i.e., change in postural positions and post-exercise recovery) might also affect the moment-to-moment cardiovascular adaptation, reflecting poor homeostasis and vulnerability to functional disturbances and diseases [50,51,52,53].

Thus, we cannot establish a cause–and–effect relationship even though we had controlled athletes’ weekly exercise volume and all participants were clinically evaluated and considered fit to carry out the present research. Therefore, we cannot answer if the impairment observed in the HG is a chronic autonomic adaptation state of HIITCE. Thus, new studies should be carried out to answer this question. Furthermore, the results cannot be extrapolated to men of different ages who practice HIITCE at an athletic level, men and women who practice HIITCE recreative of the same age group, and older adults.

## 5. Conclusions

In conclusion, this study showed that healthy athletes who practice exclusively high-intensity interval training combined with different exercises (i.e., CrossFit^®^) at rest (supine) showed reduced parasympathetic and increased sympathetic modulation, low reactivity after the postural change, and impaired HRR compared to TG. However, the cross-sectional observational design does not allow a claim that the cardiac autonomic response observed in the HG during the analysis (activity, reactivity, and reactivation) was a response that starts from a chronic autonomic adaptation state of HIITCE (i.e., characteristic of this modality). New studies should be conducted to answer the effect on cardiac autonomic function in HIITCE athletes.

## Figures and Tables

**Figure 1 ijerph-20-00634-f001:**
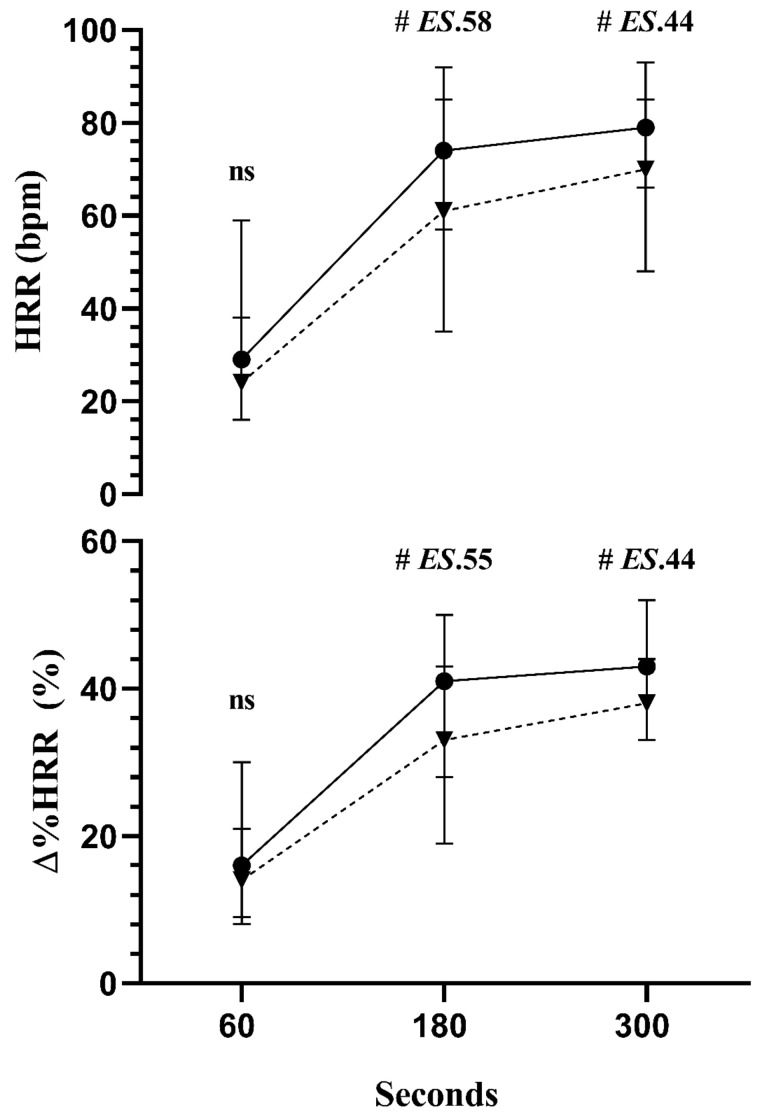
The median (25th, 75th percentiles) of HRR: heart rate recovery and Δ% HRR at 60, 180, and 300 s. The triathlon group (TG_:_ ●_;_ n = 16, solid line) and the high-intensity interval training combined with different types of exercises group (HG: ▼; n=18, dashed line) were compared by Mann–Whitney test. **#**: *p* < 0.01. ES: effect size. NS: non-significant.

**Table 1 ijerph-20-00634-t001:** Median (25th, 75th percentiles) of anthropometrical and basic physiological variables in the triathlon group (TG) and high-intensity interval training combined with different types of exercises group (HG).

Variables	TG	HG	*p*
Age (years)	32.5 (30–26)	30.5 (27–34)	0.39
BMI (kg/m^2^)	23.3 (22–24)	26 (23–27)	0.01 *
Sleep per Day (hours)	7.1 (6.4–8)	6.7 (6–7.6)	0.41
Training Volume (h/week)	17 (15–19)	7 (6–8)	0.001 *
HR_supine_ (bpm)	51 (49–55)	56.5 (51–62)	0.015 *
HR_orthostatic_ (bpm)	65 (63–72)	73 (63–78)	0.11
SBP_supine_ (mmHg)	130 (120–130)	129 (120–130)	0.11
DBP_supine_ (mmHg)	90 (80–90)	89.5 (80–90)	0.70
SBP_orthostatic_ (mmHg)	120 (112–128)	120 113–128)	0.12
DBP_orthostatic_ (mmHg)	88 (80–94)	80 (76–88)	0.17
RR_supine_ (breaths/min)	13.5 (11–16)	13.5 (10–17)	0.41
RR_orthostatic_ (breaths/min)	15 (12–17)	14.5 11–16)	0.74
HR_max_ (bpm)	185 (179–195)	185 (180–192)	0.57
Test duration (s)	620 (592–656)	588 (596–618)	0.33
VO_2 peak_ mL(kg.min)^−1^	55 (52–58)	50 (47–52)	0.001 *

***** Value statistically different (*p* < 0.05) when compared with control group by Mann–Whitney test. BMI: body mass index; HR: heart rate; SBP: systolic blood pressure; DBP: diastolic blood pressure; RR: respiratory rate; HR: heart rate; VO_2_ peak: maximal oxygen consumption.

**Table 2 ijerph-20-00634-t002:** Median (25th, 75th percentiles) of the mean RR intervals and the frequency-domain indices of the triathlon group (TG; n = 16) and the high-intensity interval training combined with different types of exercises group (HG; n = 18) in supine and orthostatic positions.

	Supine				Orthostatic			
	TG	HG	*p*	*ES*	TG	HG	*p*	*ES*
Mean RRi (ms)	1173 (967–1313)	1077 (923–1255)	0.00 *	0.46	915 (669–1033)	824 (681–1107)	0.30	0.17
Total power (ms^2^)	625 (111–5301)	1188 (205–4768)	0.10	0.28	647 (88–2379)	509 (212–1895)	0.89	0.02
LF power (ms^2^)	161 (30–2457)	346 (80–1916)	0.01 *	0.42	285 (20–1012)	248 (45–880)	0.53	0.10
HF power (ms^2^)	190 (34–1858)	292 (24–1775)	0.70	0.06	42 (6–155)	72 (9–479)	0.15	0.24
LF normalized area (%)	0.43 (0.31–0.46)	0.52 (0.47–0.74)	0.01 *	0.41	0.86 (0.75v 0.91)	0.76 (0.67–0.88)	0.11	0.12
HF normalized area (%)	0.55 (0.53–0.67)	0.47 (0.24–0.52)	0.01 *	0.12	0.13 (0.07–0.23)	0.21 (0.10–0.31)	0.13	0.31
LF/HF ratio	0.7 (0.1–2.5)	1 (0.1–14.7)	0.01 *	0.42	6 (1–35)	3 (0.6–12)	0.09	0.28

***** *p*-value statistically different (*p* ≤ 0.05) for Mann–Whitney test; ES: effect size.

**Table 3 ijerph-20-00634-t003:** Median (25th, 75th percentiles) of absolute (∆ ABS) and relative (∆ %) changes from the supine to the orthostatic positions of the frequency-domain indices of the triathlon group (TG; n=16) and the high-intensity interval training combined with different types of exercises group (HG; n = 18). Legend: * Value statistically different (*p* ≤ 0.05) when compared with triathlon group by Mann–Whitney test. LF: low frequency; HF: high frequency.

	∆ A.B.S.				∆ (%)			
	TG	HG	*p*	*ES*	TG	HG	*p*	*ES*
Total power (ms ^2^)	−119 (−661; 142)	−340 (−957; −110)	0.12	0.26	−26 (−55; 50)	−31 (−61; −21)	0.16	0.23
LF power (ms ^2^)	127 (−30; 264)	−73 (−704; 26)	0.02 *	0.38	102 (−33; 393)	−37 (−72; −48)	0.01 *	0.46
HF power (ms ^2^)	−151 (−460; −50)	−174 (−355; −48)	0.83	0.03	−80 (−86; −72)	−66 (−91; −30)	0.21	0.21
LF normalized area (%)	0.40 (0.30–0.57)	0.20 (0.00–0.40)	0.01 *	0.31	91.2 (54.5–175.7)	32 (1750–80.9)	0.01 *	0.40
HF normalized area (%)	−0.40 (−0.57; −0.30)	−0.20 (−0.40; −0.00)	0.01 *	0.07	−77.5 (−88.9; −63)	−41.5 (−71.1; −1.8)	0.01 *	0.12
LF/HF ratio	5.4 (2.8; 11)	1.6 (−0.4; 4.5)	0.01 *	0.46	863 (345;1349)	128 (2.6; 745)	0.01 *	0.50

***** *p*-value statistically different (*p* ≤ 0.05) for Mann–Whitney test; ES: effect size.

## Data Availability

The data presented in this study are available on request from the corresponding author. The data are not publicly available due to future planned analysis.

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
