# Peer review of "High-Intensity Interval Training Combined with Different Types of Exercises on Cardiac Autonomic Function. An Analytical Cross-Sectional Study in CrossFit® Athletes"

_ijerph, 2022, doi:10.3390/ijerph20010634_

Round 1

Reviewer 1 Report

1. The observation of autonomic function and regulation during Crossfit training is a novel experiment and also necessary and timely, but the lack of a control group and the cross-sectional design significantly diminish the value of this study. 

2. Another weakness is that the study was not a random sample, which makes the representativeness of the sample questionable. With such a design, a confounding factor, i.e. a difference between the two groups due to the participant's own characteristics, cannot be well excluded.

3. Another questionable point is that this study is a comparison of two training modalities that were adapted to the actual training in the experimental design? How can it be determined that the difference was caused by the training modality and not by some other factor of the training, such as the duration of the training?

4. For the above 3 reasons, the authors should be especially careful when drawing conclusions from this study. 

5. Why was this time chosen for the test when the observation time for the experiment was 48 hours after training? In the discussion section of the study, the authors state that the results observed in this study are not "an acute or chronic state of adaptation". If this is the case, should the authors explain what state was actually observed in this study? What is the theoretical and practical value of the data from this condition?

6. How were the specific times for the tests determined? Why was this choice made?

7. why do you not simply use the word CrossFit but another word to describe or refer to this training?

8. Line 41: "The autonomic function of the heart is affected by physical inactivity". This sentence should be reconsidered as the reference () indicates correlation rather than causation.

9. Lines82-84:" ... characterised by high-volume exercise with predominant oxidative demand as opposed to low-volume exercise and HIITCE on cardiac autonomic The sentence "function..." is not clear and needs to be rewritten.

10. Discussion section: Do the results of this study have any practical implications?

11. The presentation of figure 1 is unclear. What do the solid and dashed lines represent respectively? How do the markings for significance of differences in the legend compare? In short, this figure and the legend should be carefully revised.

12. the comments under Tables 2 and 3 are superfluous.

Author Response

RESPONSES TO THE REVIEWERS – R1  2022/11/18

Manuscript ID: ijerph-2027354

High-intensity interval training combined with different types of exercises on cardiac autonomic function. An analytical cross-sectional study in CrossFit ® athletes

Morlin MT, da Cruz CJG, Guimarães FER, Silva RAS, Porto LGG and Molina GE

We thank the Editor-in-Chief of the International Journal of Environmental Research and Public Health for the opportunity to resubmit our paper, and we are very grateful to the reviewers for their comments and suggestions, which importantly improved our manuscript.

We have highlighted edits to facilitate further review. We believe the responses and additions satisfactorily address the reviewers' comments and suggestions and hope you will agree. The edits have clarified several points and resulted in an improved manuscript, which we hope is ready for final acceptance.

# To the Reviewer 1    

1) The observation of autonomic function and regulation during Crossfit training is a novel experiment and also necessary and timely, but the lack of a control group and the cross-sectional design significantly diminish the value of this study.

We agree with the reviewer's comments! Thus, to highlight these limitations. We have written the study's limitations (lines from 148 to160 in the discussion section). However, we would like to draw attention to the fact that in health, whether in public health or medicine, the most systematic research is carried out in observational studies. Thus, in this type of scientific investigation, the investigators do not interfere with the phenomena under study; they only observe in a systematic and standardized way, collect and record information, data, or materials that occur spontaneously at a given moment and then proceed with their description and analysis. Therefore, the objective of this cross-sectional study was to obtain reliable data that would make it possible to generate conclusions and create new hypotheses that can be investigated with further research.

Zangirolami-Raimundo, Juliana, Echeimberg, Jorge de Oliveira, & Leone, Claudio. (2018). Research methodology topics: Cross-sectional studies. Journal of Human Growth and Development,28(3),356-360.https://dx.doi.org/10.7322/jhgd.152198

Kesmodel US. Cross-sectional studies - what are they good for? (2018). Acta Obstet Gynecol Scand.97(4):388-393. doi: 10.1111/aogs.13331.

  1. Another weakness is that the study was not a random sample, which makes the representativeness of the sample questionable. With such a design, a confounding factor, i.e., a difference between the two groups due to the participant's own characteristics, cannot be well excluded.

We agreed to the reviewer's comments, and the manuscript was adjusted in limitations (lines 149 to161 in the discussion section).

  1. Another questionable point is that this study is a comparison of two training modalities that were adapted to the actual training in the experimental design? How can it be determined that the difference was caused by the training modality and not by some other factor of the training, such as the duration of the training?

We thank the reviewer for their comments and questions. All cardiac autonomic measurements (rest, reactivity, and reactivation) were recorded using clinical protocols previously validated (PMID: 8598068 and PMID: 10536127). a) Regarding the exercise during the maximal exercise testing, there was no duration (time) difference between groups, as we showed in Table 1. b) Concerning the duration of the athletes' training, we added as a study limitation (lines from 149 to161 in the discussion section). Therefore, the groups did not present a difference in the test duration. In this manner, we controlled all possible variables and added limitations to all those we could not control. However, we would highlight that our study goes beyond the design of the laboratory, and we analyzed the participants as they presented themselves in real life to provide meaningful information on community action, with controlled and uncontrolled factors that outstrip the factors considered in clinical trials.

Roche N, Reddel HK, Agusti A, Bateman ED, Krishnan JA, Martin RJ, Papi A, Postma D, Thomas M, Brusselle G, Israel E, Rand C, Chisholm A, Price D; Respiratory Effectiveness Group. Integrating real-life studies in the global therapeutic research framework. (2013) Lancet Respir Med. 1(10):e29-30. doi: 10.1016/S2213-2600(13)70199-1.

Hohmann AA, Shear MK. Community-based intervention research: coping with the "noise" of real life in study design. 2002. Am J Psychiatry. 159(2):201-7. doi: 10.1176/appi.ajp.159.2.201.

  1. For the above 3 reasons, the authors should be especially careful when drawing conclusions from this study.

            We agreed to the reviewer's comments, and the manuscript was adjusted in limitations (lines from 149 to161 in the discussion section) and conclusion (lines from 173 to 181 in the discussion section). We made changes in the form of writing the discussion and conclusion to emphasize the study's limitations and the non-inference of autonomic alterations, but rather to draw attention to a possible autonomic alteration and the need for more robust studies for further investigation.

  1. Why was this time chosen for the test when the observation time for the experiment was 48 hours after training? In the discussion section of the study, the authors state that the results observed in this study are not "an acute or chronic state of adaptation". If this is the case, should the authors explain what state was actually observed in this study? What is the theoretical and practical value of the data from this condition?

We thank the reviewer for their comments and questions. Considering that our sample was composed of athletes, we used 48 hours of rest before the assessment of cardiac autonomic function in different conditions (activity, reactivity, and reactivation) to reduce the potential training effect on cardiac autonomic function as described in these previous studies Alansare et al. 2018, Rockholt et al. 2020. Our study design (observational cross-sectional study) does not allow us to infer that it would be a chronic adaptation associated with an acute response (activity, reactivity, and reactivation) during the experiment. Therefore, we have adjusted the manuscript as the reviewer's comment. (Lines from 128 to168 in the discussion section).   

Alansare A, Alford K, Lee S, Church T, Jung HC. The Effects of High-Intensity Interval Training vs. Moderate-Intensity Continuous Training on Heart Rate Variability in Physically Inactive Adults. Int J Environ Res Public Health. 2018 Jul 17;15(7):1508. doi: 10.3390/ijerph15071508. PMID: 30018242; PMCID: PMC6069078.

Rockholt B.K,  Grosicki G.J, Flatt A (2020). A. Resting and exercise-related heart rate responses to high intensity interval training in women: A pilot study . Journal of Physical Education and Sport 20(5):2760-2764. Doi:10.7752/jpes.2020.05375

  1. How were the specific times for the tests determined? Why was this choice made?

We thank the reviewer for their question! We performed all tests between 8:00 and 10:00 am to control the impact of circadian rhythm on athletes' cardiac autonomic function (PMID: 18779437). Yet, all athletes performed the tests throughout the maintenance phase; thus, none were in a specific training or pre-competition stage. Indeed, testing the athletes during the maintenance phase and considering a cross-sectional observational approach was necessary because the athletes could attend this study without changing their work activities (lines from 102 to 104 in the Materials and Methods section). So, we chose the current approach to increase the control of confounder (circadian rhythm and periodization training cycle) on cardiac autonomic function.

  1. why do you not simply use the word CrossFit but another word to describe or refer to this training?

            We thank the reviewer for their question! In a recent discussion of exercise concepts, CrossFit® has been recognized as high-intensity interval training combined with different types of exercise (HIITCE), which is why we did not use this term in the present study (lines 62 to 65 in the introduction).

Ide, B.N., Silvatti, A.P., Marocolo, M., Santos, C.P.C., Silva, B.V.C., Oranchuk, D.J., Mota, G.R., (2022). Is There Any Non-functional Training? A Conceptual Review. Front Sports Act Living 3, 803366. Doi: 10.3389/fspor.2021.803366

  1. Line 41: "The autonomic function of the heart is affected by physical inactivity". This sentence should be reconsidered as the reference () indicates correlation rather than causation.

As requested, we have adjusted the text. Adjusted in the manuscript (lines from 36 to 38)

The cardiac autonomic function is associated with physical inactivity, and its impairment is a predictor of all-cause and cardiovascular disease mortality

  1. Lines82-84:" ... characterized by high-volume exercise with predominant oxidative demand as opposed to low-volume exercise and HIITCE on cardiac autonomic The sentence "function..." is not clear and needs to be rewritten.

Thank the reviewer for their comment. We have adjusted the manuscript (lines 66 to 78 in the introduction).

  1. Discussion section: Do the results of this study have any practical implications?

We thank the reviewer for their comments and questions! Yes, you can find it in the discussion section, lines 128 to 148.

Lauer MS. Autonomic function and prognosis. Cleveland Clinic Journal of Medicine. 2009 Apr;76 Suppl 2:S18-22. DOI: 10.3949/ccjm.76.s2.04. PMID: 19376976.

Saffitz JE. Sympathetic neural activity and the pathogenesis of sudden cardiac death. Heart Rhythm. 2008 Jan;5(1):140-1. doi: 10.1016/j.hrthm.2007.10.039. Epub 2007 Nov 1. PMID: 18083070.

DeBlauw JA, Drake NB, Kurtz BK, Crawford DA, Carper MJ, Wakeman A, Heinrich KM. High-Intensity Functional Training Guided by Individualized Heart Rate Variability Results in Similar Health and Fitness Improvements as Predetermined Training with Less Effort. J Funct Morphol Kinesiol. 2021 Dec 13;6(4):102. doi: 10.3390/jfmk6040102. PMID: 34940511; PMCID: PMC8705715.

  1. The presentation of figure 1 is unclear. What do the solid and dashed lines represent respectively? How do the markings for significance of differences in the legend compare? In short, this figure and the legend should be carefully revised.

As requested, we have revised and adjusted Figure 1.  Figure 1 was adjusted in lines 5 to 8.

  1. the comments under Tables 2 and 3 are superfluous.

As requested, we have adjusted and removed superfluous legends in tables 2 and 3. Legend in table 2 was adjusted in line 242 and Table 3.

Reviewer 2 Report

First of all, I would like to thank the authors for the opportunity to review their submitted manuscript "High-intensity interval training combined with different types of exercises on cardiac autonomic function. An analytical cross-sectional study in CrossFit ® athletes". In this well-designed study, the authors showed that CrossFit® athletes, compared with triathlon athletes, exhibit sympathetic activation at rest, as well as low reactivity after posture changes and impaired heart rate recovery after exercise. These data are interesting and contain new scientific results, however, when reviewing, I had a number of questions that I would like to get answers from the authors.

1.     The authors' data on cardiac autonomic function in CrossFit athletes (higher degree of sympathetic activation compared to triathlon athletes) may have different interpretations. First, it is possible that these results may indicate the presence of the overreaching and overtraining syndrome (at least in the initial stage). This may be due to the less developed control of training intensity among CrossFit athletes. An example of a recent study (1) was that modulating HIFT exercise intensity based on individual HRV status was more effective than standard HIFT (which was also used in this study). Therefore, the question arises of how the state of overtraining or overreaching was excluded in sportsmen. At the same time, were authors guided only by subjective symptoms (feelings of fatigue, sleep disturbances, apathy, or restlessness) or was an additional examination carried out? What was the examination of the cardiologist - only a survey of athletes, or did he evaluate some objective signs of overtraining or overreaching (2)?

2.     Secondly, perhaps this result may be a characteristic feature of CrossFit® programs. We can agree with the opinion of the authors of the article that the design of the article (lack of a sedentary control group, cross-sectional nature of the study, lack of prospective observation) does not allow answering this question. However, in the discussion could attempt to provide literature data on the possible impact of CrossFit® programs on autonomic balance (similar to similar comparisons, such as high-intensity interval training and Moderate-Intensity Continuous Training (3))

3.     Finally, the question arises of the correctness of the comparison of heart rate recovery in the CardioPulmonary exercise test. In the triathlon group, the athletes were trained for a similar load, in contrast to the CrossFit athletes. Therefore, differences in heart rate recovery after the test may be due not to differences in the state of autonomic balance, but to varying degrees of muscle adaptation to this type of load.

References

1.     DeBlauw JA, Drake NB, Kurtz BK, Crawford DA, Carper MJ, Wakeman A, Heinrich KM. High-Intensity Functional Training Guided by Individualized Heart Rate Variability Results in Similar Health and Fitness Improvements as Predetermined Training with Less Effort. J Funct Morphol Kinesiol. 2021 Dec 13;6(4):102. doi: 10.3390/jfmk6040102.

2.     Carrard J, Rigort AC, Appenzeller-Herzog C, Colledge F, Königstein K, Hinrichs T, Schmidt-Trucksäss A. Diagnosing Overtraining Syndrome: A Scoping Review. Sports Health. 2022 Sep-Oct;14(5):665-673. doi: 10.1177/19417381211044739.

3.     van Biljon A, McKune AJ, DuBose KD, Kolanisi U, Semple SJ. Short-Term High-Intensity Interval Training Is Superior to Moderate-Intensity Continuous Training in Improving Cardiac Autonomic Function in Children. Cardiology. 2018;141(1):1-8. doi: 10.1159/000492457.

Author Response

RESPONSES TO THE REVIEWERS – R1  2022/11/18

Manuscript ID: ijerph-2027354

High-intensity interval training combined with different types of exercises on cardiac autonomic function. An analytical cross-sectional study in CrossFit ® athletes

Morlin MT, da Cruz CJG, Guimarães FER, Silva RAS, Porto LGG and Molina GE

We thank the Editor-in-Chief of the International Journal of Environmental Research and Public Health for the opportunity to resubmit our paper, and we are very grateful to the reviewers for their comments and suggestions, which importantly improved our manuscript.

We have highlighted edits to facilitate further review. We believe the responses and additions satisfactorily address the reviewers' comments and suggestions and hope you will agree. The edits have clarified several points and resulted in an improved manuscript, which we hope is ready for final acceptance.

# To the Reviewer 2

First of all, I would like to thank the authors for the opportunity to review their submitted manuscript, "High-intensity interval training combined with different types of exercises on cardiac autonomic function. An analytical cross-sectional study in CrossFit ® athletes". In this well-designed study, the authors showed that CrossFit® athletes, compared with triathlon athletes, exhibit sympathetic activation at rest, as well as low reactivity after posture changes and impaired heart rate recovery after exercise. These data are interesting and contain new scientific results, however, when reviewing, I had a number of questions that I would like to get answers from the authors.

1.The authors' data on cardiac autonomic function in CrossFit athletes (higher degree of sympathetic activation compared to triathlon athletes) may have different interpretations. First, it is possible that these results may indicate the presence of the overreaching and overtraining syndrome (at least in the initial stage). This may be due to the less developed control of training intensity among CrossFit athletes. An example of a recent study (1) was that modulating HIFT exercise intensity based on individual HRV status was more effective than standard HIFT (which was also used in this study). Therefore, the question arises of how the state of overtraining or overreaching was excluded in sportsmen. At the same time, were authors guided only by subjective symptoms (feelings of fatigue, sleep disturbances, apathy, or restlessness) or was an additional examination carried out? What was the examination of the cardiologist - only a survey of athletes, or did he evaluate some objective signs of overtraining or overreaching (2)?

  1. DeBlauw JA, Drake NB, Kurtz BK, Crawford DA, Carper MJ, Wakeman A, Heinrich KM. High-Intensity Functional Training Guided by Individualized Heart Rate Variability Results in Similar Health and Fitness Improvements as Predetermined Training with Less Effort. J Funct Morphol Kinesiol. 2021 Dec 13;6(4):102. doi: 10.3390/jfmk6040102. PMID: 34940511; PMCID: PMC8705715.

  2. Carrard J, Rigort AC, Appenzeller-Herzog C, Colledge F, Königstein K, Hinrichs T, Schmidt-Trucksäss A. Diagnosing Overtraining Syndrome: A Scoping Review. Sports Health. 2022 Sep-Oct;14(5):665-673. doi: 10.1177/19417381211044739. Epub 2021 Sep 9. PMID: 34496702; PMCID: PMC9460078.

            We are grateful for the review comments and suggestions and for the opportunity to explain our methodological approach. Thank you! First, we have added the suggested reference to our manuscript, improving our manuscript (references 69 -70). Regarding the objective signs of overtraining or overreaching, in this study, all subjects underwent clinical-cardiological evaluation by a cardiologist before assessing heart rate variability. In this evaluation, the ECG was used and analyzed with clinical caution. Only after medical clearance the individuals underwent HRV assessment according to the protocol established in the laboratory (Porto & Junqueira, 2009).

According to study 2, specific hormones, neurotransmitters, and metabolites, as well as psychological, electrocardiographic, electroencephalographic, and immunological patterns, were identified as potential diagnostic for OTS, reflecting its multisystemic nature. Hence, in the present study, subjective symptoms and electrocardiograms were analyzed. Thus, we have raised the question of the possibility of overtraining or overreaching (at least in the initial stage) in the discussion (lines 68 to 77).

Porto, L.G.G., Junqueira, L.F., (2009). Comparison of time-domain short-term heart interval variability analysis using a wrist-worn heart rate monitor and the conventional electrocardiogram. Pacing Clin Electrophysiol 32, 43–51. doi: 10.1111/j.1540-8159.2009.02175.x

2.Secondly, perhaps this result may be a characteristic feature of CrossFit® programs. We can agree with the opinion of the authors of the article that the design of the article (lack of a sedentary control group, cross-sectional nature of the study, lack of prospective observation) does not allow answering this question. However, in the discussion could attempt to provide literature data on the possible impact of CrossFit® programs on autonomic balance (similar to similar comparisons, such as high-intensity interval training and Moderate-Intensity Continuous Training (3)

We thank the reviewer for their suggestion. The manuscript was adjusted. (Lines from 63 to 68) in the discussion section.

3.Finally, the question arises of the correctness of the comparison of heart rate recovery in the CardioPulmonary exercise test. In the triathlon group, the athletes were trained for a similar load, in contrast to the CrossFit athletes. Therefore, differences in heart rate recovery after the test may be due not to differences in the state of autonomic balance, but to varying degrees of muscle adaptation to this type of load.

We thank the reviewer for their comments and questions. All cardiac autonomic measurements (rest, reactivity, and reactivation) were recorded using clinical protocols previously validated (PMID: 8598068 and PMID: 10536127). Regarding the exercise during the maximal exercise testing, there was no duration (time) difference between groups, as we showed in Table 1. Yet, the exercise testing and recovery cool-down procedure were employed considering their technical feasibility, reproducibility, and usual application in the clinical setting (PMID: 26888648). However, we have highlighted in the discussion section that differences between groups during the recovery phase (180 and 300s) might be an important functional characteristic of both exercise modalities (lines from 112 to 118 in the discussion section) and could be related to the clearance of metabolites and catecholamines released during exercise (Gallina et al., 2011) which may be related to muscle adaptation which can change the cardiac autonomic modulation.

Gallina, S., Di Mauro, M., D’Amico, M.A., D’Angelo, E., Sablone, A., Di Fonso, A., Bascelli, A., Izzicupo, P., Di Baldassarre, A., (2011). Salivary chromogranin A, but not α-amylase, correlates with cardiovascular parameters during high-intensity exercise. Clin Endocrinol (Oxf) 75, 747–752. Doi: 10.1111/j.1365-2265.2011.04143.x

Reviewer 3 Report

Title: High-intensity interval training combined with different types of exercises on cardiac autonomic function. An analytical cross-sectional study in CrossFit athletes

Article Type: Article

Summary

The present study compared the cardiac autonomic function at rest, its reactivity, and reactivation following the maximal exercise testing of two groups of male athletes (HIITCE versus endurance). Heart rate variability and heart rate recovery of CrossFit and triathlon athletes were recorded in several times. The findings indicated that CrossFit athletes at rest showed sympathetic increase and reduction of parasympathetic activities, low reactivity after the postural change, and impairment HRR, which may indicate a possible sympathetic increase in the CrossFit athletes.

Evaluation

The topic of this study is interesting for publication in the Journal. The design for the study is appropriate to answer the research questions, and the paper is well written. However, there are some points should be addressed by the authors, in order to improve the quality of the manuscript.  

Suggestions

-Please add mean age ± SD of participants to the abstract.

-Please speak a little bit about the research methods and study design in the abstract.

-In the title and purpose of your research, you have said that you want to investigate the effect of high-intensity interval training on the mentioned indicators, so why do you have a group as an endurance athletes? Wouldn't it be better if you had a control group?

- Considering that many researches have been done in this field, it is better to mention more researches in the introduction section.

- Please talk more about the research gap and the reason for doing the research in the introduction section.

-In the method section, you said, inclusion was “at least one year of regular exercise routine”, is this period enough? What was your reason for selecting this?

-Please add the exclusion criteria too.

-How the sample size was calculated?

- Please speak more about the application of the results as well as the limitations of the research in the discussion and conclusion section.

Author Response

RESPONSES TO THE REVIEWERS – R1 2022/11/18

Manuscript ID: ijerph-2027354

High-intensity interval training combined with different types of exercises on cardiac autonomic function. An analytical cross-sectional study in CrossFit ® athletes

Morlin MT, da Cruz CJG, Guimarães FER, Silva RAS, Porto LGG and Molina GE

We thank the Editor-in-Chief of the International Journal of Environmental Research and Public Health for the opportunity to resubmit our paper, and we are very grateful to the reviewers for their comments and suggestions, which importantly improved our manuscript.

We have highlighted edits to facilitate further review. We believe the responses and additions satisfactorily address the reviewers' comments and suggestions and hope you will agree. The edits have clarified several points and resulted in an improved manuscript, which we hope is ready for final acceptance.

# To the Reviewer 3

Suggestions

- Please add mean age ± SD of participants to the abstract

The authors' guidelines establish a limit of 250 words in the abstract. Therefore, considering the review request, we have adjusted the abstract to avoid exceeding the limit of 250 words.

It has been adjusted in the manuscript line 17-33.

-Please speak a little bit about the research methods and study design in the abstract.

As requested, we have adjusted as reviewer's comments, and it has been adjusted in the manuscript line from 18 to 30.

-In the title and purpose of your research, you have said that you want to investigate the effect of high-intensity interval training on the mentioned indicators, so why do you have a group as an endurance athletes? Wouldn't it be better if you had a control group?

Thank you for questioning and for the opportunity to explain our methodological approach.

It was a methodological decision that considered the well-established positive effect induced by aerobic training on cardiac autonomic function (Alansare et al., 2018; Cabral-Santos et al., 2016; Furlan et al., 1993; Kiss et al., 2016; Silva et al., 2015). On the other hand, the lack of studies on HIITCE and the cardiac autonomic function raised the hypothesis that HIITCE can improve the cardiac autonomic function (rest, reactivity, and reactivation) as aerobic training. Therefore, we decided to compare HIITCE with a modality that we know can positively improve cardiac autonomic function from the point of view clinical and functional.

Alansare, A., Alford, K., Lee, S., Church, T., Jung, H.C.,(2018). The Effects of High-Intensity Interval Training vs. Moderate-Intensity Continuous Training on Heart Rate Variability in Physically Inactive Adults. Int J Environ Res Public Health 15. doi: 10.3390/ijerph15071508

Cabral-Santos, C., Giacon, T.R., Campos, E.Z., Gerosa-Neto, J., Rodrigues, B., Vanderlei, L.C.M., Lira, F.S., (2016). Impact of High-intensity Intermittent and Moderate-intensity Continuous Exercise on Autonomic Modulation in Young Men. Int J Sports Med 37, 431–435. doi: 10.1055/s-0042-100292

Furlan, R., Piazza, S., Dell’Orto, S., Gentile, E., Cerutti, S., Pagani, M., Malliani, A. (1993). Early and late effects of exercise and athletic training on neural mechanisms controlling heart rate. Cardiovasc. Res. 27, 482–488. doi: 10.1093/cvr/27.3.482

Kiss, O., Sydó, N., Vargha, P., Vágó, H., Czimbalmos, C., Édes, E., Zima, E., Apponyi, G., Merkely, G., Sydó, T., Becker, D., Allison, T.G., Merkely, B., (2016). Detailed heart rate variability analysis in athletes. Clin Auton Res 26, 245–252. doi: 10.1007/s10286-016-0360-z

Silva, V.P. da, Oliveira, N.A. de, Silveira, H., Mello, R.G.T., Deslandes, A.C., (2015). Heart Rate Variability Indexes as a Marker of Chronic Adaptation in Athletes: A Systematic Review. Annals of Noninvasive Electrocardiology 20, 108–118. doi: 10.1111/anec.12237

- Considering that many researches have been done in this field, it is better to mention more researches in the introduction section.

We thank the reviewer for their comments. As requested, it has been adjusted in the manuscript (lines 66 to 72).

- Please talk more about the research gap and the reason for doing the research in the introduction section.

We thank the reviewer again! As requested, it has been adjusted in the manuscript (lines 66-78).

--In the method section, you said, inclusion was "at least one year of regular exercise routine", is this period enough? What was your reason for selecting this?

We thank the reviewer for their question! We decided on this approach based on previous studies, which highlighted that a high degree of performance, competition, and training in any sport is achieved only after a regular basis on three days per week for more than one year of scheduled training and continuous progression over training (Petrik, 2014; Meier N et al., 2021; Mangine GT and Seay TR, 2022). In addition, similarly to Meier N et al. (2021) and Peña J et al. (2021), our participants exhibited an average time experience of 3.3 and 4.0 years for CrossFit® and triathlon, respectively.

Mangine GT, Seay TR. Quantifying CrossFit®: Potential solutions for monitoring multimodal workloads and identifying training targets. Front Sports Act Living. 2022 Oct 14;4:949429. doi: 10.3389/fspor.2022.949429. PMID: 36311217; PMCID: PMC9613943.

Peña J, Moreno-Doutres D, Peña I, Chulvi-Medrano I, Ortegón A, Aguilera-Castells J, Buscà B. Predicting the Unknown and the Unknowable. Are Anthropometric Measures and Fitness Profile Associated with the Outcome of a Simulated CrossFit® Competition? Int J Environ Res Public Health. 2021 Apr 1;18(7):3692. doi: 10.3390/ijerph18073692. PMID: 33916215; PMCID: PMC8037316.

Meier N, Rabel S, Schmidt A. Determination of a CrossFit® Benchmark Performance Profile. Sports (Basel). 2021 Jun 2;9(6):80. doi: 10.3390/sports9060080. PMID: 34199523; PMCID: PMC8228530.

Petrik, M. CrossFit Powerworkouts: Intensivtraining für Kraft & Ausdauer; BLV Buchverlag: Munich, Germany, 2014.

-Please add the exclusion criteria too.

As requested, we have adjusted the manuscript (lines 144 to146).

-How the sample size was calculated?

Thank you very much for the question. We performed a nonparametric statistical test for the comparison of 2 independent groups, two tails, an effect size of 0.9 and a power of 0.8 with an error probability of 0.05, and the sample size was 44 subjects, using the Gpower 3.1.9.7 software. Therefore, we collected 50 subjects. However, due to sample mortality, 34 athletes were included.

Therefore, considering the sample mortality, we calculated the observed power (OP). As we wrote in the text (lines 212 to 214 in the materials and method section): "The observed power (OP) was calculated by post hoc power analyses for each analysis variable using G*Power 3.1.9.7 for Windows software. Post hoc power analyses often make sense after a study has already been conducted. In post hoc analyses, 1 – β is computed as a function of α, the population effect size parameter, and the sample size(s) used in a study. It thus becomes possible to assess whether a published statistical test had a fair chance of rejecting an incorrect H0. Post hoc analyses, like a priori analyses, require an H1 effect size specification for the underlying population. Post hoc power analyses should not be confused with so-called retrospective power analyses, in which the effect size is estimated from sample data and used to calculate the observed power, a sample estimate of the true power.

At any alpha level, increased sample sizes always yield greater statistical test power, and a potential problem then turns into excessive power. By "excessive," it means that increasing the sample size implies that smaller and smaller effects will be perceived as statistically significant until almost every effect is significant in very large samples. The researcher must always be aware that the sample size can impact the statistical test, making it insensitive (with small samples) or overly sensitive (with very large samples).

The relationships between alpha, sample size, effect size, and power are very complicated, and many guidance references are available. Cohen examines power for most tests of statistical inference and guides acceptable levels of power, suggesting that studies should be designed to achieve alpha levels of at least 0.05 with power levels of 80%. To achieve such levels of power, the three factors – alpha, sample size, and effect size – must be considered simultaneously.

HAIR, Joseph F. Multivariate data analysis. 2009.

FIELD, Andy; MILES, Jeremy; FIELD, Zoë. Discovering statistics using R. 2012.

- Please speak more about the application of the results as well as the limitations of the research in the discussion and conclusion section.

We thank the reviewer for their suggestion. You can find it on lines 127 to 167 in the discussion section and from lines 172 to 180 in the conclusion section.

Round 2

Reviewer 1 Report

I thank the authors for their prompt response and clarification of my questions.

Reviewer 2 Report

The authors significantly corrected the article, answered all my questions and comments, as well as other reviewers. I don't have any other questions.